# DriveLM: Driving with Graph Visual Question Answering

Paper ID NA

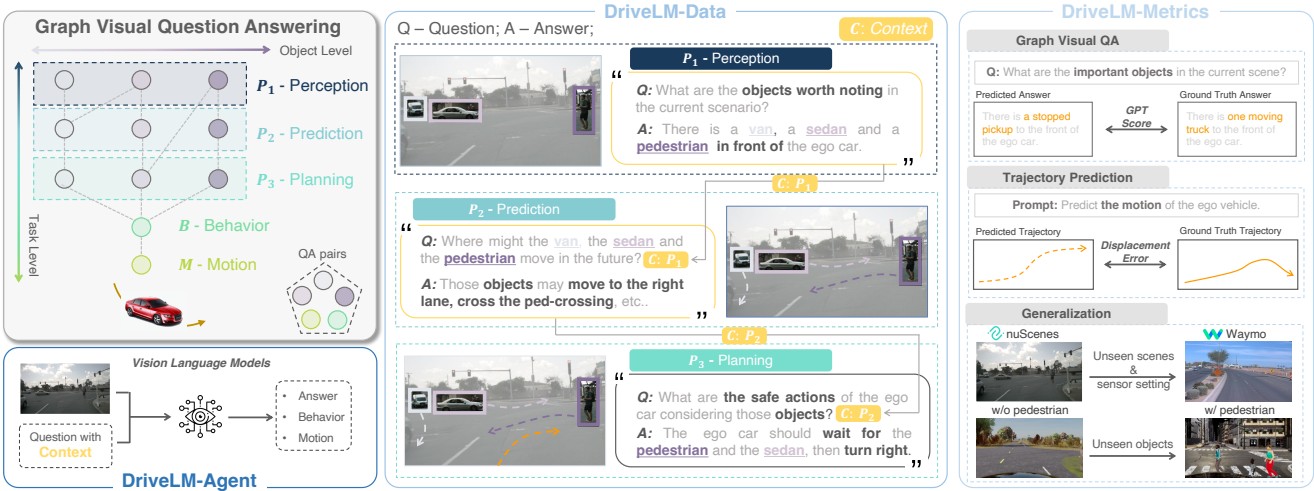

Figure 1. We present **DriveLM**: A new task, dataset, metrics, and baseline for end-to-end autonomous driving. Inspired by [4], DriveLM considers **Graph Visual Question Answering (GVQA)**, where question-answer pairs are interconnected via logical dependencies at the object-level, *i.e.*, interactions between object pairs, and the task-level, *e.g.*, perception → prediction → planning → behavior (discretized action described in natural language) → motion (continuous trajectory). We propose **DriveLM-Data** for training **DriveLM-Agent**, a baseline for GVQA. We validate its effectiveness using the **DriveLM-Metrics** on challenging settings requiring zero-shot generalization.

## Abstract

*We study how vision-language models (VLMs) trained on web-scale data can be integrated into end-to-end driving systems to boost generalization and enable interactivity with human users. While recent approaches adapt VLMs to driving via single-round visual question answering (VQA), human drivers reason about decisions in multiple steps. Starting from the localization of key objects, humans estimate object interactions before taking actions. The key insight is that with our proposed task, Graph VQA, where we model graph-structured reasoning through perception, prediction and planning question-answer pairs, we obtain a suitable proxy task to mimic the human reasoning process. We instantiate datasets (DriveLM-Data) built upon nuScenes and CARLA, and propose a VLM-based baseline approach (DriveLM-Agent) for jointly performing Graph VQA and end-to-end driving. The experiments demonstrate that Graph VQA provides a simple, principled framework for reasoning about a driving scene, and DriveLM-Data provides a challenging benchmark for this task. Our DriveLM-Agent baseline performs end-to-end autonomous driving competitively in comparison to state-of-the-art driving-specific architectures. Notably, its benefits are pronounced when it is evaluated zero-shot on unseen objects or sensor configurations. We hope this work can be the starting point to shed new light on how to apply VLMs for autonomous driving. To facilitate future research, all code, data, and models are available to the public.*

## 1. Introduction

Current Autonomous Driving (AD) stacks are still lacking crucial capabilities [4, 5]. One key requirement is generalization, which involves the ability to handle unseen scenarios or unfamiliar objects. A secondary requirement pertains to the interaction of these models with humans, highlighted for example by EU regulations that mandate explainability

---

*Equal contribution. †Equal co-advising.

in deployment [1]. Furthermore, unlike today's AD models, humans do not navigate based on geometrically precise bird's-eye view (BEV) representations [6, 13, 16]. Instead, humans implicitly perform object-centric perception, prediction, and planning (which we refer to as $P_{1-3}$): a rough identification and localization of key objects, followed by reasoning about their possible movement and aggregation of this information into a driving action [22, 27].

Simultaneously, another field has been forging ahead: Vision-Language Models (VLMs) [17, 19, 30, 34]. These models have several strengths. First, they hold a base understanding of the world from internet-scale data that could potentially facilitate generalization for planning in AD. In fact, this sort of generalization has already been achieved by VLMs for simpler robotics tasks [9, 35]. Second, the use of language representations as an input and output offers a platform for human-friendly interaction with these models, unlike bounding boxes or trajectories that are more common to current methods [7, 12, 18, 25]. Finally, VLMs are able to make decisions in multiple steps linked by logical reasoning [2, 8, 31–33, 35]. Importantly, even though they reason in multiple separate steps, VLMs are end-to-end differentiable architectures, a characteristic that is highly desirable for autonomous driving [4].

Recent work towards enabling the application of VLMs to AD systems falls into two categories: scene-level or single object-level Visual Question Answering (VQA). Scene-level VQA refers to the task of describing the driving behavior by one or two supporting reasons, *e.g.*, "The car is moving into the right lane because it is safe to do so." [14, 15]. Single object-level VQA formulates the understanding of the ego vehicle's response to a single object by a chain of QAs in the form of "what-which-where-how-why", *e.g.*, "The ego vehicle stops because there is a pedestrian in a white shirt crossing the intersection in front of the ego vehicle and it does not want to crash into the pedestrian." [21, 24, 26]. Unfortunately, neither of these paradigms provides a suitable proxy task to mimic the $P_{1-3}$ reasoning process in humans, who consider multiple objects and reason about each in multiple steps. Therefore, in this paper, we propose a new task, along with corresponding datasets and a baseline model architecture (Fig. 1).

**Task. Graph Visual Question Answering (GVQA)** involves formulating $P_{1-3}$ reasoning as a series of question-answer pairs (QAs) in a directed graph. Its key difference to the aforementioned VQA tasks for AD is the availability of logical dependencies between QAs which can be used to guide the answering process. GVQA also encompasses questions regarding behavior and motion planning, with dedicated metrics (details in Section 2).

**Datasets. DriveLM-nuScenes** consist of annotated QAs, arranged in a graph, linking images with driving behavior through logical reasoning. In comparison to existing benchmarks, they provide significantly more text annotations per frame (Fig. 2). We pair these training datasets with challenging test data for evaluating zero-shot generalization.

**Model. DriveLM-Agent** employs a trajectory tokenizer that can be applied to any general VLM [17, 19, 23, 34], coupled with a graph prompting scheme that models logical dependencies as context inputs for VLMs. The result is a simple, elegant methodology to effectively repurpose VLMs for end-to-end AD.

Our experiments provide encouraging results. We find that GVQA on DriveLM is a challenging task, where current methods obtain moderate scores and better modeling of logical dependencies is likely necessary to achieve strong QA performance. Even so, DriveLM-Agent already performs competitively to state-of-the-art driving-specific models [13] when tested in the open-loop planning setting, despite its task-agnostic and generalist architecture. Furthermore, employing a graph structure improves zero-shot generalization, enabling DriveLM-Agent to correctly handle novel objects unseen during training or deployment on the Waymo dataset [28] after training only on nuScenes [3] data. From these results, we believe that improving GVQA holds great potential towards building autonomous driving agents with strong generalization.

## 2. DriveLM: Task, Data, Metrics

Human drivers usually decompose their decision-making process into distinct stages that follow a logical progression which encompasses the identification and localization of key objects, their possible future action and interaction, and ego planning based on all this information [10, 20]. This inspires us to propose the GVQA as the critical ingredient of DriveLM, which serves as a suitable proxy task to mimic the human reasoning process. Within this section, we illustrate the formulation of the GVQA task (Section 2.1) and introduce DriveLM-Data (Section 2.2) to exemplify the instantiation of GVQA using prominent driving datasets.

### 2.1. DriveLM-Task: GVQA

We organize all the Question Answer pairs (QAs) for an image frame into a graph structure, denoted by $G = (V, E)$. $V$ stands for the set of vertices, where each vertex represents a QA pair $v = (q, a)$ associated with one or more key objects in the scenario. The key difference between GVQA and ordinary VQA is that the QAs in GVQA have logical dependencies, which we formulate as the edges between the vertices. $E \subseteq V \times V$, is a set of directed edges, where each edge $e = (v_p, v_c)$ connects the parent QA and the child QA. We formulate the edge set $E$ by incorporating two dimensions: object-level and task-level edges. At the object level, we construct the logical edges $e \in E$ to represent the impact of interactions between different objects. For example, the

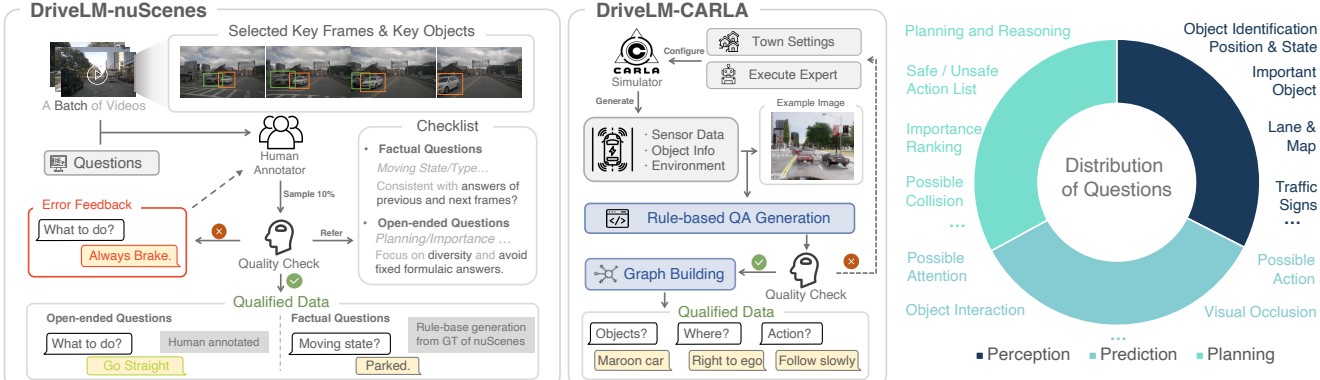

Figure 2. **(Left) Annotation Pipeline:** In DriveLM-nuScenes, we adopt a semi-rule-based QA labeling pipeline, where both the ground truth annotation in nuScenes/OpenLane-V2 and feedback from human annotators are used. A critical part of our pipeline is the multi-round quality check, which guarantees high data quality at reasonable costs. In DriveLM-CARLA, we meet the same standards while exploiting a fully rule-based QA labeling pipeline instead. **(Right) Question Distribution:** The questions in our dataset cover various specific aspects of driving tasks, most of which are annotated by human annotators, making this a suitable proxy for human-like driving reasoning.

planning QA node for the sedan is influenced by the perception QA node of the pedestrian in the illustration from Fig. 1 (center). At the task-level, we establish the logical edges $e \in E$ to capture the logical chain of different reasoning stages:

- **Perception** ($P_1$): identification, description, and localization of key objects in the current scene.
- **Prediction** ($P_2$): estimation of possible action/interaction of key objects based on perception results.
- **Planning** ($P_3$): possible safe actions of the ego vehicle.
- **Behavior** ($B$): classification of driving decision.
- **Motion** ($M$): waypoints of ego vehicle future trajectory.

The concepts of perception, prediction, and planning ($P_{1-3}$) are similar to those in end-to-end AD [4], while the concepts of motion and behavior are based on the ego vehicle future trajectory. Specifically, we define the motion $M$ as the ego vehicle future trajectory, which is a set of $N$ points with coordinates $(x, y)$ in bird's-eye view (BEV), denoted as $M = \{(x_0, y_0), (x_1, y_1), ..., (x_N, y_N)\}$. Each point is the offset between the future position and the current position by a fixed time interval. Then, the distance for $x, y$ at each time interval is computed as:

$$\{x, y\}_{\text{dist}} = \{(\delta_{x,1}, \delta_{y,1}), ..., (\delta_{x,N}, \delta_{y,N})\}, \quad (1)$$

where $\delta_{x,i} = x_i - x_{i-1}$ and $\delta_{y,i} = y_i - y_{i-1}$, for $i = 1, 2, \ldots, N$. The goal of the behavior representation is to serve as an interface from $P_{1-3}$ to $M$. To obtain a behavior representation, we map the mean of $x_{\text{dist}}$ and $y_{\text{dist}}$ to one of the predefined bins, where each bin corresponds to a category in either speed or steering. These are denoted as $B_{sp}$ and $B_{st}$ respectively. In this work, we consider 5 bins:

$$B_{sp} \in \{\texttt{fast}_2, \texttt{fast}_1, \texttt{moderate}, \texttt{slow}_1, \texttt{slow}_2\},$$

$$B_{st} \in \{\texttt{left}_2, \texttt{left}_1, \texttt{straight}, \texttt{right}_1, \texttt{right}_2\},$$

where the number in the subscript indicates the intensity. The combination of the speed and steering categories for a trajectory form its behavior category as $B = (B_{sp}, B_{st})$. While we use a simple definition of $B$ as a starting point for research on driving with VLMs, we note that our formulation supports the incorporation of more abstract behaviors such as a lane changes or overtaking.

## 2.2. DriveLM-Data

We introduce DriveLM-nuScenes to provide QAs with the graph structure defined in Section 2.1,

**DriveLM-nuScenes.** We divide the annotation process into three steps: selecting key frames from video clips, choosing key objects within these key frames, and subsequently annotating the frame-level $P_{1-3}$ QAs for these key objects. A portion of the Perception QAs are generated from the nuScenes [3] and OpenLane-V2 [29] ground truth, while the remaining QAs are manually annotated. As we manually annotate the vast majority of data in DriveLM-nuScenes, quality is particularly crucial for this portion. When annotating, we conduct multiple rounds of rigorous quality checks. In each round, we categorize the data into different batches and inspect ten percent of the data in each batch. If the qualification rate of manually annotated data in this ten percent does not meet expectations, we request the annotators to re-label all data in the batch. In Fig. 2 (left), we showcase an example of the QA annotation pipeline, where all questions undergo quality checks according to our standards. As a result, DriveLM-nuScenes stands out from previously proposed datasets with its larger scale, greater comprehensiveness, and more complex structure. These QAs cover various aspects of the driving process, ranging from perception and prediction to planning, providing a comprehensive understanding of autonomous driving scenarios as shown in Fig. 2 (right).

## 3. Experiments

In this section, we present our experimental results that aim to address the following research questions: (1) How can VLMs be effectively repurposed for end-to-end autonomous driving? (2) Can VLMs for driving generalize when evaluated with unseen sensor setups;

**Setup.** We now briefly overview the key implementation details for the two settings used in our experiments (additional details are provided in the supplementary material). All fine-tuning is implemented with LoRA [11]. On DriveLM-nuScenes, we finetune BLIP-2 on the `train` split for 10 epochs. We use a batch size of 2 for each GPU, and the entire training process spans approximately 7 hours with 8 V100 GPUs.

### 3.1. VLMs for End-to-End Driving

In our first experiment, we aim to assess the ability of VLMs to perform open-loop planning on DriveLM-nuScenes. In particular, we investigate the impact of the context provided to the behavior and motion stages. Given sensor data (and in the case of VLM methods, a text input), the model is required to predict the ego-vehicle future trajectory in the form of waypoints.

**Baselines.** As a reference for the difficulty of the task, we provide a simple **Command Mean** baseline. Each frame in nuScenes is associated with one of 3 commands, 'turn left', 'turn right', or 'go straight'. We output the mean of all trajectories in the training set whose command matches the current test frame command. Further, we compare our approach to the current state-of-the-art on nuScenes, UniAD [13]. Besides the author-released checkpoint, which requires video inputs, we train a single-frame version ('**UniAD-Single**') for a fair comparison to our single-frame VLMs. Finally, **BLIP-RT-2** denotes BLIP-2 [17] fine-tuned on DriveLM-Data with the trajectory tokenization scheme. This acts as an indicator for the performance when using an identical network architecture as DriveLM-Agent, but no context inputs or VQA training data.

**DriveLM-Agent.** We consider 3 variants of DriveLM-Agent incorporating our proposed changes in steps: (1) a 2-stage version that predicts behavior and then motion (as described in Section 2.1), but without any $P_{1-3}$ context for behavior prediction ('None'); (2) a 'Chain' version that builds the $P_{1-3}$ graph, but only passes the final node ($P_3$) to the behavior stage; (3) the full model ('Graph') that uses all QAs from $P_{1-3}$ as context for $B$.

**Results.** We show the results for the methods listed above in Table 1. Among the baselines, BLIP-RT-2 is unable to match UniAD-Single (though both methods perform well relative to Command Mean). This shows that the single-stage approach without any reasoning is unable to compete

| Method | Behavior Context | Motion Context | Behavior ($B$) | | | Motion ($M$) | |
|---|---|---|---|---|---|---|---|
| | | | Acc. ↑ | Speed ↑ | Steer ↑ | ADE ↓ | Col. ↓ |
| Command Mean | - | - | - | - | - | 4.57 | 5.72 |
| UniAD-Single | - | - | - | - | - | 1.80 | 2.62 |
| BLIP-RT-2 | - | - | - | - | - | 2.63 | 2.77 |
| DriveLM-Agent | None | $B$ | **61.45** | **72.20** | **84.73** | **1.39** | **1.67** |
| | Chain | $B$ | 50.43 | 60.32 | 75.34 | 2.07 | 2.08 |
| | Graph | $B$ | 57.49 | 69.89 | 80.63 | 1.74 | 1.89 |
| *UniAD* [13] | - | - | - | - | - | *0.80* | *0.17* |

Table 1. **Open-loop Planning on DriveLM-nuScenes.** Using Behavior ($B$) as context for Motion ($M$) enables end-to-end driving with VLMs on par with UniAD-Single, a state-of-the-art driving-specific architecture.

| Method | Behavior Context | Motion Context | Behavior ($B$) | | | Motion ($M$) | |
|---|---|---|---|---|---|---|---|
| | | | Acc. ↑ | Speed ↑ | Steer ↑ | ADE ↓ | FDE ↓ |
| Command Mean | - | - | - | - | - | 7.98 | 11.41 |
| UniAD-Single | - | - | - | - | - | 4.16 | 9.31 |
| BLIP-RT-2 | - | - | - | - | - | 2.78 | 6.47 |
| DriveLM-Agent | None | $B$ | 35.70 | 43.90 | 65.20 | 2.76 | 6.59 |
| | Chain | $B$ | 34.62 | 41.28 | 64.55 | 2.85 | 6.89 |
| | Graph | $B$ | **39.73** | **54.29** | **70.35** | **2.63** | **6.17** |

Table 2. **Zero-shot Generalization across Sensor Configurations.** Results on 1k randomly sampled frames from the Waymo `val` set after training on DriveLM-nuScenes. DriveLM-Agent outperforms UniAD-Single and benefits from graph context.

with the prior state-of-the-art on nuScenes. However, the proposed DriveLM-Agent, which predicts behavior as an intermediate step for motion, provides a significant boost in performance, surpassing UniAD-Single. This indicates that with the appropriate prompting, VLMs can be surprisingly competitive for end-to-end driving. Interestingly, in the experimental setting of Table 1 which does not involve generalization, the Chain and Graph versions of DriveLM-Agent do not provide any further advantage over no context. Further, single-frame VLMs fall short in comparison to the privileged video-based UniAD model, indicating that VLMs with video inputs may be necessary for this task.

### 3.2. Generalization Across Sensor Configurations

As a more challenging setting for evaluating the models from Section 3.1, we now apply them without any further training to a new domain: the Waymo dataset [28]. Waymo's sensor setup does not include a rear camera, so we drop this input from UniAD-Single. The VLM methods only use the front view and do not require any adaptation.

**Results.** As shown in Table 2, UniAD-Single does not cope well with the new sensor configuration, and drops below BLIP-RT-2 in performance. The multi-stage approach of DriveLM-Agent provides further improvements. In particular, the accuracy of speed predictions rises from 43.90 with no context to 54.29 with the full graph. On the other hand, the chain approach does not provide sufficient useful information, with a speed accuracy of only 41.28.

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
