# OpenReview forum: "DriveLM: Driving with Graph Visual Question Answering"
_thecvf.com/CVPR/2024/Workshop/VLADR — VLADR 2024 Oral_

### Official Review · Reviewer_zd7f · 2024-04-17
**Generally good-quality paper with some minor concerns.**

**Rating:** 8
**Confidence:** 4

**Review:**

Summary:

This work introduces a new framework which integrates vision-language models into end-to-end autonomous driving systems. Through a task called Graph Visual Question Answering, this task mimics human decision-making processes in driving by using a series of interconnected question-answer pairs that logically depend on each other across object and task levels. These levels encompass perception, prediction, planning, behavior, and motion.

Strengths:

The introduction of GVQA as a task for autonomous driving systems is innovative. Graph decomposition allows sufficient explainability in the context of end-to-end autonomous driving.

The DriveLM-Data contains rich annotations and provides a robust platform for the training and validation of similar models.

The proposed framework has better zero-shot generalization capability across different sensor configurations.

Weaknesses:

The framework is single-framed, however, many SOTA methods on NuScene datasets, including UniAD, show that multi-frame methods are generally better since they could take account into the historical information. The complexity of the proposed framework (graph-based VQA) might grow exponentially in multi-frame context, make it challenging to scale up to video-based framework.

"Interestingly, in the experimental setting of Table 1 which does not involve generalization, the Chain and Graph versions of DriveLM-Agent do not provide any further advantage over no context." Does the paper have any explanation for this?

Suggestions:

It will be great to have some discussion on how the proposed work's method is different comparing with other existing LLM-powered driving models (e.g., DriveGPT4, GPT-Driver, etc.).

---

### Decision · Program_Chairs · 2024-04-22

Accept (Oral)